# Simultaneous solving high-resolution structures of various enzymes from human kidney microsomes

Meinan Lyu, Chih-Chia Su ⬧, Masaru Miyagi, Edward W Yu ⬧

**The ability to investigate tissues and organs through an integrated systems biology approach has been thought to be unobtainable in the field of structural biology, where the techniques mainly focus on a particular biomacromolecule of interest. Here we report the use of cryo-electron microscopy (cryo-EM) to define the composition of a raw human kidney microsomal lysate. We simultaneously identify and solve cryo-EM structures of four distinct kidney enzymes whose functions have been linked to protein biosynthesis and quality control, biosynthesis of retinoic acid, gluconeogenesis and glycolysis, and the regulation and metabolism of amino acids. Interestingly, all four of these enzymes are directly linked to cellular processes that, when disrupted, can contribute to the onset and progression of diabetes. This work underscores the potential of cryo-EM to facilitate tissue and organ proteomics at the atomic level.**

## Introduction

The kidney is an essential organ that helps maintain whole-body homeostasis. Humans and vertebrates have two kidneys located on the left and right sides of the retroperitoneal space. They are responsible for detoxification by excreting into the urine a variety of waste products, including toxic substances and those generated via normal metabolic processes. They also maintain physiologic water and salt levels and are capable of regulating acid–base balance, extracellular fluid volume, and blood pressure. In addition, the kidneys secrete several hormones, including renin, erythropoietin, and calcitriol, to sustain bodily functions. Transcriptome analysis suggests that 75% of all human proteins are produced in the kidney, whereas 413 genes have elevated expression in this organ when compared with all other organs and tissues (Uhlen et al, 2015). This makes the kidney an important organ to study, both at the gross and mechanistic levels.

We recently developed a cryo-electron microscopy (cryo-EM) methodology termed "Build and Retrieve" (BaR) (Su et al, 2021). This is an iterative process capable of performing in silico purification and sorting of images from a large, heterogeneous dataset containing several different proteins and biomacromolecules. We rationalized that the BaR methodology can be used to elucidate systems' structural proteomics of human tissue and organ samples and employed this method to study human kidney microsomes because the kidney plays a major role in bodily functions. In addition, chronic kidney diseases have been recognized as a leading health issue worldwide, with ~7 million patients required to have kidney replacement surgery in 2010 (Lv & Zhang, 2019). CKDs are also a leading cause of death in the United States. It is estimated that ~37 million US adults suffer from chronic kidney diseases (Kidney Disease Statistics for the United States, 2021).

In this study, we enriched proteins from raw lysate of human kidney microsomes using size exclusion chromatography. We then used single-particle cryo-EM imaging to simultaneously identify and solve cryo-EM structures of four different kidney enzymes involved in protein biosynthesis, biosynthesis of retinoic acid, gluconeogenesis and glycolysis, and regulation and metabolism of amino acids. Intriguingly, all of these enzymes have a direct connection with the chronic health condition diabetes.

## Results

Enrichment of proteins from the kidney microsomal raw lysate using size exclusion chromatography allowed us to obtain two major peaks located at 100–200 kD and 250–650 kD (Fig S1). We separately collected single-particle images of these two peaks using cryo-EM and processed these datasets using the BaR methodology (Su et al, 2021). Several iterative rounds of 2D classification permitted us to sort the images into different protein classes (Fig S2). The BaR approach was then used which allowed us to determine cryo-EM structures of four different kidney enzymes with resolutions ranging between 2.62 and 2.88 Å (Table S1). These enzymes were identified as the heterodimeric complex glucosidase II (GANAB), retinaldehyde dehydrogenase 1 (ALDH1A1), fructose-bisphosphate aldolase (FPA), and betaine-homocysteine methyltransferase (BHMT).

We also used proteomic analysis to study the composition of the two human kidney microsomal lysate peaks. We confirmed that

---

Department of Pharmacology, Case Western Reserve University School of Medicine, Cleveland, OH, USA

Correspondence: edward.w.yu@case.edu

 

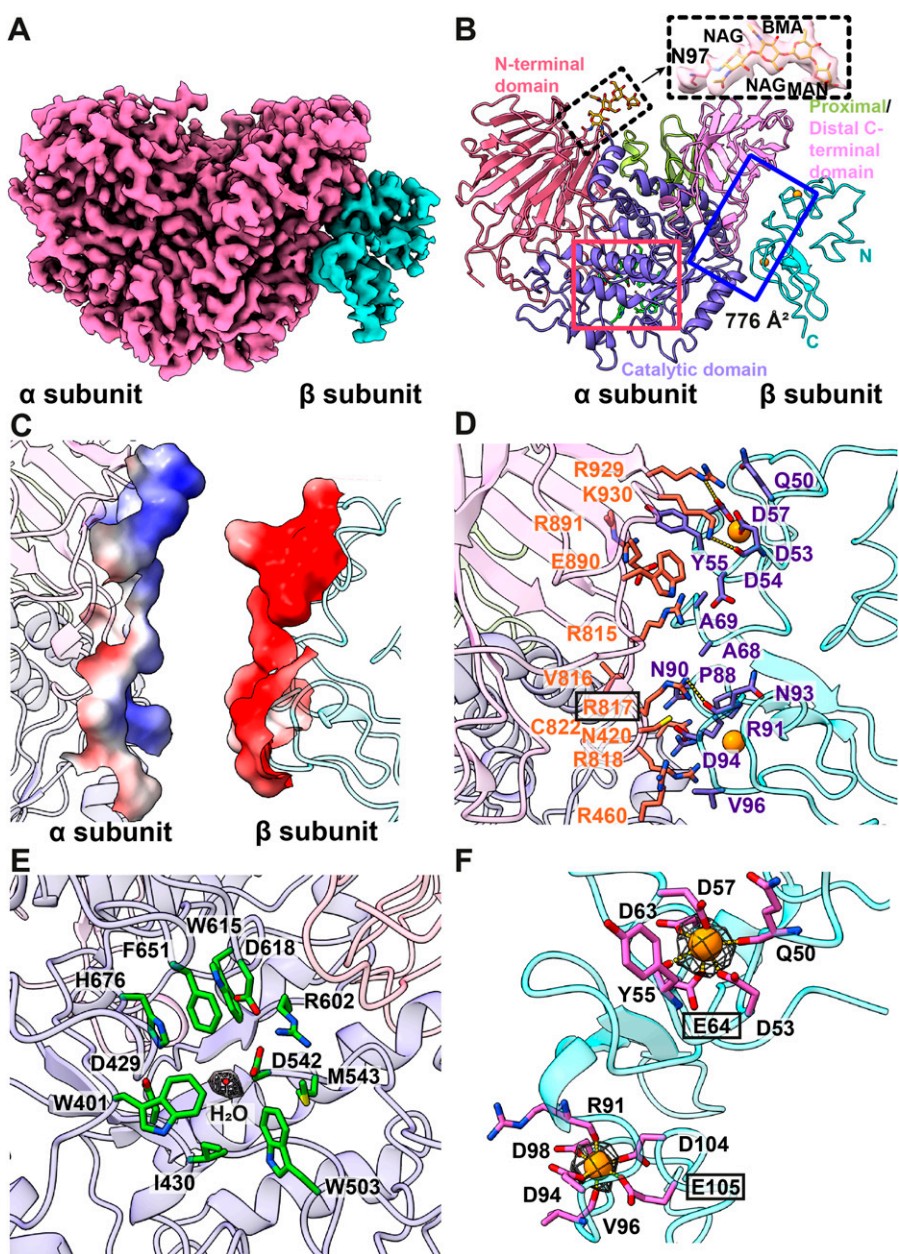

**Figure 1. Cryo-EM structure of human kidney GANAB.**
**(A)** Cryo-EM density map of the heterodimeric GANAB enzyme, consisting of the α- and β-subunits. The α- and β-subunits are colored pink and cyan. **(B)** Ribbon diagram of the 2.88-Å resolution structure of GANAB. The N-terminal, catalytic, proximal C-terminal, and distal C-terminal domains of the α-subunit are colored pink, slate, lemon, and light pink, respectively. The β-subunit is colored cyan. The two bound Ca²⁺ ion at the β-subunit are in orange spheres. A N-linked glycosylation was observed at residue N97. The density of the NAG-NAG-BMA-MAN glycan chain is in wheat mesh. **(C)** Electrostatic surface potential of GANAB. Surface representation of the protein–protein interface between the α- and β-subunits are colored by charge (red, negative −15 kT/e; blue, positive +15 kT/e). The secondary structural elements of the α- and β-subunits are colored light pink and cyan. **(D)** Important residues for subunit–subunit interaction. Residues involved in heterodimerization are highlighted with orange and slate sticks. The two bound Ca²⁺ ions at the β-subunit are in orange spheres. The dotted lines indicate hydrogen bonds. **(E)** Active site of GANAB. Residues W401, D429, I430, W503, D542, M543, R602, W615, D618, F651, and H676 are in green sticks. A mutation on residue R817 can cause autosomal-dominant polycystic liver disease. This residue is highlighted with a black rectangle. The secondary structural elements of the α- and β-subunits are colored slate and pink. Density corresponding to the water molecule, which is highlighted with a red sphere, is in gray mesh. **(F)** Ca²⁺-binding sites of the β-subunit. The two bound Ca²⁺ ions are in orange spheres. Residues involved in Ca²⁺ binding are in magenta sticks. Mutations on residues E64 and E105 result in inactivating the function of GANAB. These two residues are highlighted with black rectangles. Densities corresponding to the bound Ca²⁺ ions are in gray meshes.

each peak contains more than 400 proteins. The 10 most abundant proteins based upon the protein identification score of each peak are listed in Table S2. The presence of the four enzymes identified by BaR in our sample was also confirmed by this proteomic analysis Table S2).

## Structure of the heterodimeric complex glucosidase II

Human GANAB plays a critical role in protein folding and quality control. GANAB is a heterodimeric enzyme complex composed of a catalytic α-subunit active enzyme and a non-catalytic β-subunit accessory protein. This enzyme catalyzes the hydrolysis of peptide-bound oligosaccharides and participates in the quality control

assessment of glycoprotein folding (Porath et al, 2016). As such, GANAB is recognized as an important member of the endoplasmic reticulum quality control machinery. It should be noted that mutations in GANAB have been found to cause the life-threatening autosomal-dominant polycystic kidney disease (Porath et al, 2016).

We collected a total of 440,504 single-particle projections for this class of images. The BaR protocol allowed us to identify this protein as the human GANAB enzyme. We then determined the first structure of this human GANAB protein to a resolution of 2.88 Å (Figs 1 and S3 and Table S1).

Our cryo-EM structure indicates that human kidney GANAB is composed of one α-subunit and one β-subunit (Fig 1A), which is in good agreement with the crystal structure of the homologous

mouse enzyme α-glucosidase II (Caputo et al, 2016). Protein sequence alignment shows that these two enzymes share 96% protein sequence similarity. Superimposition of the cryo-EM structure of the human enzyme to the x-ray structure of the mouse enzyme (PDB ID: 5F0E) (Caputo et al, 2016) gives rise to a root-mean-square deviation of 0.52 Å (for 850 Cα atoms).

The α-subunit of human kidney GANAB can be divided into an N-terminal domain, a catalytic domain, and a C-terminal domain (Fig 1B). The final cryo-EM structure includes 878 amino acids (residues 33–186 and 221–944) for the α-subunit of this enzyme. The N-terminal domain is mostly β-stranded, except residues 45–52 which form an α-helix at the N-terminal end. The catalytic domain adopts a fold of the GH31 family (Henrissat, 1991) which contains a mixture of the α-helical and β-sheet secondary structural elements. The C-terminal domain can be subdivided into proximal and distal C-terminal domains, where these two subdomains create 7-stranded and 10-stranded β-barrels, respectively. Interestingly, an extra, elongated density was observed at residue N97, indicating that this residue is likely to be glycosylated. The shape of this elongated density is compatible with a long glycan chain in the form of NAG-NAG-BMA-MAN with NAG, BMA, and MAN coded for N-acetylglucosamine, β-D-mannose, and α-D-mannose, respectively (Fig 1B). This observation is indeed in good agreement with results from a biochemical study that determined that human GANAB is glycosylated (Martiniuk et al, 1985). The x-ray structure of mouse α-glucosidase II also depicts that this enzyme is glycosylated at residue N97 with the same elongated glycan chain (Caputo et al, 2016).

The β-subunit of GANAB is quite disordered. We could only include residues 18–117 in our final model. It appears that most of the secondary structural elements present as unstructured random loops (Fig 1B).

Intriguingly, it appears that the α- and β-subunits of GANAB specifically interact with each other. The surface of the α-subunit is highly electropositive, whereas the β-subunit is highly electronegative at the α/β subunit–subunit interface (Fig 1C). Thus, the nature of subunit–subunit interaction is mostly governed by electrostatic interactions. Residues participating in creating this 776 $Å^2$ α/β subunit interface are depicted in Fig 1D. Specifically, the positively charged residues R460, R815, R817, R818, R891, R929, and K930 of the α-subunit and negatively charged residues D53, D54, D57, and D94 of the β-subunit are specifically engaged in these electrostatic interactions. It should be noted that R929 and K930 of the α-subunit contribute to two hydrogen bonds with D57 and D53 of the β-subunit. R817 of the α-subunit also forms a hydrogen bond with the backbone oxygen of N93 of the β-subunit to strengthen subunit–subunit interaction (Fig 1D). In addition, R815, R818, and R929 of the α-subunit and D54, D94, and D53 of the β-subunit participate in creating three salt bridges to stabilize binding.

These residues at the subunit–subunit interface could be very important for the function of GANAB, where a mutation of one or more of these residues could lead to devastating illnesses. For example, a missense mutation R817W has been identified in three patients with autosomal-dominant polycystic liver disease (Porath et al, 2016). Based on our cryo-EM structure, R817 is critical for stabilizing the interaction between α and β subunits as this α-subunit residue directly contacts N93 of the β-subunit to form a hydrogen bond (Fig 1D).

The active site of the α-subunit of GANAB is lined with several hydrophobic and charged residues, including W401, D429, I430, W503, D542, M543, R602, W615, D618, F651, and H676 (Fig 1E). The corresponding residues in mouse GANAB have been found to interact with substrates (Caputo et al, 2016). In our cryo-EM structure, we observed an extra density occupying this site, in which we modeled this density with a water molecule.

Notably, two hexacoordinated $Ca^{2+}$ ions are found within the β-subunit (Fig 1B and F). One $Ca^{2+}$ is ligated by the side chains of Q50, D53, Y55, D57, D63, and E64 (Fig 1F), while the other $Ca^{2+}$ is connected to the side chains of D94, D98, D104, and E105 and also the backbone oxygens of R91 and V96 (Fig 1F). A study of the homologous α-glucosidase II enzyme from Schizosaccharomyces pombe using alanine scanning mutagenesis indicated that mutations of residues E73 and E114 of the β-subunit inactivated its function. This work underscored the importance of these glutamates to coordinate with $Ca^{2+}$ ions. The two corresponding residues in the human kidney GANAB enzyme are E64 and E105 which are also responsible for anchoring $Ca^{2+}$ ions (Fig 1F).

### Structure of retinaldehyde dehydrogenase 1

The ALDH1A1 enzyme plays a major role in retinoic acid biosynthesis. Retinoic acid is an important signaling molecule that specifically interacts with the retinoic acid receptor. This signaling event controls several vital developmental processes, including neurogenesis, cardiogenesis, and development of the eye, forelimb bud, and foregut (Duester, 2008). ALDH1A1 is highly expressed in the kidney and liver, and mutations of this enzyme have been associated with a number of human diseases, including cancer, Parkinson's disease, and obesity (Ziouzenkova et al, 2007; Wey et al, 2012; Tomita et al, 2016).

We collected a total of 466,604 single-particle cryo-EM projections for this class of protein images. Based on these projections, the BaR methodology allowed us to construct a high-resolution cryo-EM map. Subsequently, we were able to identify this protein as the ALDH1A1 enzyme and resolve its structure to a resolution of 2.84 Å (Figs 2 and S4 and Table S1).

ALDH1A1 assembles as a tetrameric enzyme (subunits 1–4) with each molecule of ALDH1A1 containing 501 amino acids. The final de novo model of this enzyme includes residues 17–501 of subunits 1 and 2 and residues 9–501 of subunits 3 and 4. Like other crystal structures of ALDHs (Moore et al, 1998; Morgan & Hurley, 2015), each ALDH1A1 subunit consists of a cofactor-binding domain, a catalytic domain, and an oligomerization domain.

Surprisingly, subunits 1 and 2 of ALDH1A1 do not directly interact with each other (Fig 2A and B). The tetramer is assembled in such a way that subunits 1 and 3 form an extensive subunit–subunit interface of 2,814 $Å^2$ in area (Fig 2B and C). Specifically, S444 and Y481 of subunit 1 interact with K490 and H157 of subunit 3, respectively, to form hydrogen bonds to anchor these two subunits. In addition, the backbone oxygen of F469 and R476 of subunit 1 contact E488 and the backbone oxygen of V489 of subunit 3, respectively, to create two additional hydrogen bonds at this interface. In addition, E480 of subunit 1 forms a salt bridge with R143 of subunit 3 to further strengthen subunit–subunit interactions. This interface is more or less identical to the interface formed between subunits 2 and 4.

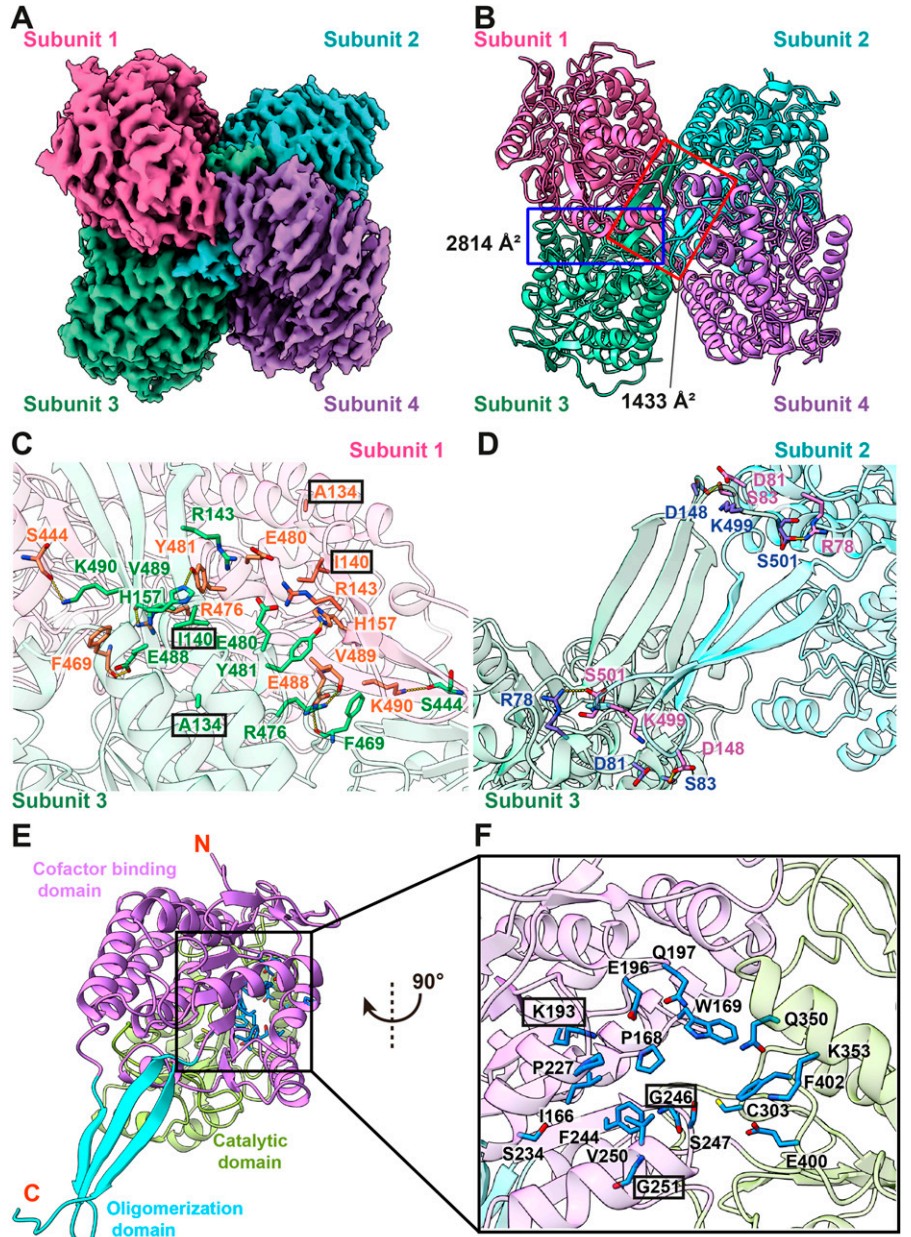

**Figure 2. Cryo-EM structure of human kidney ALDH1A1.**
**(A)** Cryo-EM density map of tetrameric ALDH1A1. The four protomers of ALDH1A1 are colored magenta, cyan, green, and violet, respectively. **(B)** Ribbon diagram of the 2.20-Å resolution structure of tetrameric ALDH1A1. The four protomers are colored magenta, cyan, green, and violet, respectively. **(C)** Subunits 1 and 3 interactions. Residues involved in subunit–subunit interactions are highlighted with orange (for subunit 1) and green (for subunit 3) sticks. Mutations on residues A134 and I140 result in congenital heart disease. These residues are highlighted with black rectangles. The secondary structural elements of the subunits 1 and 3 are colored pink and green. **(D)** Subunits 2 and 3 interactions. Residues involved in subunit–subunit interactions are highlighted with pink (for subunit 2) and slate (for subunit 3) sticks. The secondary structural elements of the subunits 2 and 3 are colored cyan and green. **(E)** A monomer of ALDH1A1. The secondary structural elements of the cofactor-binding, catalytic, and oligomerization domains of ALDH1A1 are colored pink, green, and cyan, respectively. Conserved residues at the NAD$^+$-binding site are in cyan sticks. **(F)** The NAD$^+$-binding site. Residues involved in NAD$^+$ binding are highlighted with blue sticks. Mutations on residues K193, G246, and G251 have been reported to significantly change the activity of ALDH1A1. These residues are highlighted with black rectangles. The secondary structural elements of the cofactor-binding, catalytic, and oligomerization domains are colored pink, green, and cyan, respectively.

Subunits 2 and 3 also create a second subunit–subunit interface of 1,433 Å$^2$ to strengthen tetrameric oligomerization (Fig 2B and D). Notably, R78, S83, D148, and S501 of subunit 2 contact S501, D148, S83, and R78 of subunit 3, respectively, to form four hydrogen bonds at the subunit–subunit interface. In addition, D81 and K499 of subunit 2 interact with K499 and D81 of subunit 3 via salt bridges to reinforce subunit–subunit interactions. The interactions at the interface between subunits 1 and 4 are more or less the same as those found between subunits 2 and 3.

These subunit–subunit interface residues are likely important for the function of this enzyme. Two missense mutations, A151S and I157T, have been identified for the human ALDH1A2 enzyme (Christy & Doss, 2015). These two mutations are strongly associated with

congenital heart disease. The corresponding two amino acids in ALDH1A1 are the interface residues A134 and I140 (Fig 2C), where they may be critical for the tetrameric oligomerization of this enzyme.

The cofactor of ALDH1A1 is nicotinamide adenine dinucleotide (NAD$^+$). NAD$^+$ is important as it can accept electrons to reduce to NADH to maintain redox homeostasis. This cofactor-binding site exhibits a typical $\beta\alpha\beta$ motif of the Rossmann fold (Rossmann et al, 1974; Wierenga et al, 1986). Several conserved residues, including I166, K193, E196, Q350, E400, and F402, are found within the NAD$^+$-binding site (Fig 2E and F). These residues are engaged in housing the NAD$^+$ cofactor. In addition to these conserved residues, P168, W169, Q197, P227, S234, F244, G246, S247, G251,

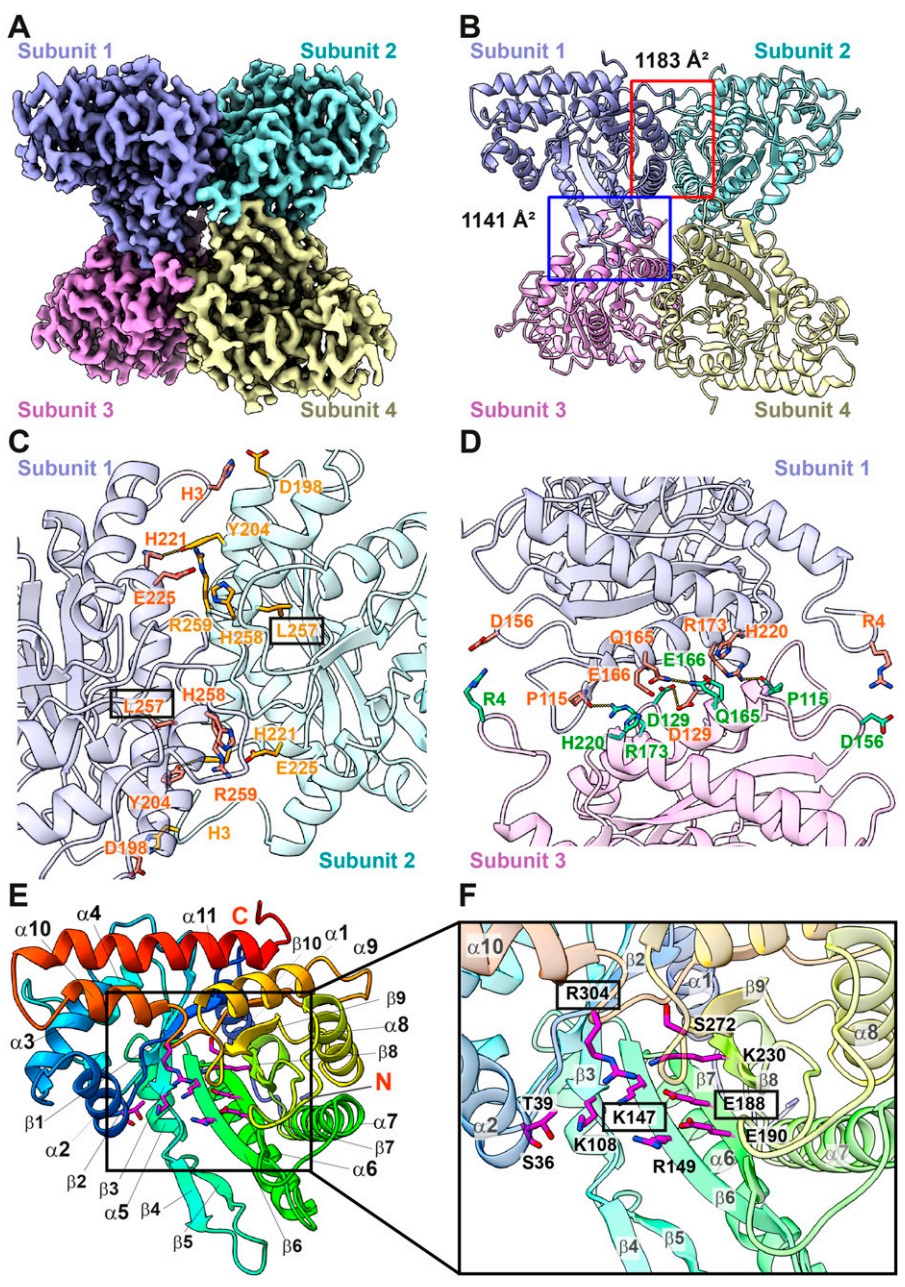

**Figure 3. Cryo-EM structure of human kidney FPA.**
**(A)** Cryo-EM density map of tetrameric FPA. The four protomers of FPA are colored slate, cyan, magenta, and yellow, respectively. **(B)** Ribbon diagram of the 2.80-Å resolution structure of tetrameric FPA. The four protomers are colored slate, cyan, magenta, and yellow, respectively. The interface between subunits 1 and 2 covers an area of 1,183 Å², whereas the interface between subunits 1 and 3 possesses an area of 1,141 Å². **(C)** Composition of the interface between subunits 1 and 2. Residues involved in subunit–subunit interaction are in orange (for subunit 1) and yellow (for subunit 2) sticks. The secondary structural elements of subunit 1 and subunit 2 are colored slate and cyan. **(D)** Composition of the interface between subunits 1 and 3. Residues involved in subunit–subunit interaction are in orange (for subunit 1) and green (for subunit 3) sticks. A mutation on residue L257 causes hereditary fructose intolerance. This residue is highlighted with a black rectangle. The secondary structural elements of subunit 1 and subunit 3 are colored slate and pink. **(E)** A monomer of FPA. The secondary structural elements are colored using a rainbow style (blue, N-terminus; red, C-terminus). The charged and polar residues lining the wall of the β-barrel are in magenta sticks. **(F)** The substrate-binding site. The charged and polar residues S36, T39, K108, K147, R149, E188, E190, K230, S272, and R304 are engaged in contacting sugar substrates. These residues are highlighted as magenta sticks. Mutations on residues K147, E188, and R304 also cause hereditary fructose intolerance. These residues are highlighted with black rectangles. The secondary structural elements of FPA are colored using a rainbow style (blue, N-terminus; red, C-terminus).

C303, and K353 are involved in NAD⁺ binding (Fig 2E and F). Our high-resolution cryo-EM map indicates that there is no bound NAD⁺ in our structure.

ALDH1A1 has been found to effectively increase NADH levels and promote tumor growth. Residues involved in creating the NAD⁺-binding site are presumed to be critical for the function of this enzyme. Indeed, it has been observed that cells harboring a mutation of the conserved interface residue K193 (K193Q or K193R) of ALDH1A1 are much less tumorigenic when compared with cells carrying the WT ALDH1A1 enzyme (Liu et al, 2021). Furthermore, another mutagenesis study indicated that the activity of ALDH1A1 was significantly reduced when two glycine residues located at the

NAD⁺-binding site, G246 and G251, were replaced by alanines (Wang et al, 2017), probably because of the effect of steric hindrance.

## Structure of FPA

FPA, often referred to as aldolase, is a key enzyme for gluconeogenesis and glycolysis. It catalyzes the reversible reaction of converting fructose 1,6-bisphosphate into dihydroxyacetone phosphate and glyceraldehyde-3-phosphate (Rutter, 1964). This enzyme is multifunctional including the capability of being engaged in a number of non-enzymatic moonlighting functions. It is able to bind various proteins that interfere with different processes such

as cellular scaffolding, signaling, transcription, and motility (Ahn et al, 1994; Wang et al, 1997; Rangarajan et al, 2010; Merkulova et al, 2011; Ritterson Lew & Tolan, 2013). Because of the multifunctional nature of these FPA enzymes, they are attractive targets for vaccine development and other molecular-based therapies to combat diseases such as cancer and pathogenic infections.

We detected a total of 715,231 single-particle images for FPA. Based on the cryo-EM map, we identified that this protein is the FPA enzyme. We then determined its cryo-EM structure to a resolution of 2.80 Å (Figs 3 and S5 and Table S1).

Kidney FPA is tetrameric in oligomerization (Fig 3A and B). Each subunit of FPA features a frame-like structure composed of 11 α-helices. Each frame-like structure also harbors a β-barrel containing 10 β-strands at the core. Thus, the tetrameric FPA enzyme carries four β-barrels that are all encircled by α-helices. The four FPA subunits anchor each other via four subunit–subunit interfaces within the tetramer (Fig 3A and B). It appears that the interface formed by subunits 1 and 2 is identical to that formed between subunits 3 and 4, whereas the interface between subunits 1 and 3 is the same as that found between subunits 2 and 4. Overall, the cryo-EM structure of human kidney FPA is in good agreement with the previously determined crystal structures of the human and rabbit FPA enzymes (Sygusch et al, 1987; Blom & Sygusch, 1997; Dalby et al, 1999; St-Jean et al, 2005).

At the interface between subunits 1 and 2, Y204 and H221 of subunit 1 specifically link to H221 and Y204 of subunit 2, respectively, to generate two hydrogen bonds. Also, H3 of subunit 1 interacts with D198 of subunit 2 to form a salt bridge, whereas E225 of subunit 1 interacts electrostatically with H258 and R259 of subunit 2 to strengthen subunit–subunit interactions (Fig 3C).

Likewise, within the interface of subunits 1 and 3, D129 and Q165 of subunit 1 associate with D129 and Q165 of subunit 3 to generate two hydrogen bonds. In addition, the backbone oxygen of P115 and R173 of subunit 1 interact with R173 and the backbone oxygen of P115 of subunit 3 to create hydrogen bonds at this interface. Four salt bridges are formed by R4, D156, E166, and H220 of subunit 1 in cooperation with D156, R4, H220, and E166 of subunit 3, further stabilizing interactions between these two subunits (Fig 3D).

Hereditary fructose intolerance is an autosomal recessive disease caused by the catalytic deficiency of FPA. Several missense mutations have been identified that are directly linked to this disease (Tolan, 1995). One of these missense mutants is the conversion of L257 to a proline (Tolan, 1995). This residue is located at the subunit–subunit interface to strengthen oligomerization (Fig 3C).

Each β-barrel is populated with charged and polar amino acids (Fig 3E and F). These residues are likely important for substrate interaction. It has been observed that the charged and polar residues S36, T39, K108, K147, R149, E188, E190, K230, S272, and R304 in human FPA are engaged in contacting the sugar fructose 1,6-bisphosphate (Dalby et al, 1999).

Several studies have shown that the glutamate and lysine residues within the substrate-binding site are critical for the function of FPA enzymes (Lai et al, 1974; Lobb et al, 1975; Hartman & Brown, 1976; Gupta et al, 1993; Morris & Tolan, 1993, 1994). Notably, a patient with hereditary fructose intolerance was found to have a six-nucleotide deletion in exon 6. This deletion perturbs the position and orientation of the corresponding K147 and E188 residues of human FPA at the active site (Fig 3F) (Santamaria et al, 1999). In addition, a separate study depicted that R304 (Fig 3F) changed to a tryptophan at this catalytic site, giving rise to hereditary fructose intolerance (Tolan, 1995). This missense mutation probably diminishes the binding of substrates and leads to this disease.

## Structure of BHMT

BHMT, also known as betaine-homocysteine S-methyltransferase, is a Zn$^{2+}$-dependent metalloenzyme that catalyzes the transfer of a methyl group from trimethylglycine and a proton from homocysteine to produce dimethylglycine and methionine (Pajares & Pérez-Sala, 2006). This enzyme is highly expressed in the kidney and liver and contributes to the regulation of cysteine levels along with the metabolism of glycine, serine, threonine, and methionine. Mutations of the BHMT gene can significantly affect the metabolism of homocysteine, leading to several illnesses, including vascular disease, schizophrenia, and spina bifida (Heil et al, 2000; Morin et al, 2003; Ohnishi et al, 2019).

We obtained 1,067,645 single-particle counts for this protein class in our cryo-EM dataset, enabling us to reveal its identity as the BHMT enzyme (Figs 4 and S6 and Table S1). We then solved the cryo-EM structure of this human kidney enzyme to a resolution of 2.62 Å.

Similar to previously determined crystal structures of BHMT enzymes (Evans et al, 2002; González et al, 2004), the overall architecture of kidney BHMT presents as a dimer of dimers (Fig 4A and B). Within this tetramer, subunits 1 and 2 tightly interact with each other to create an extensive dimer interface of 3,342 Å$^2$ (Fig 4B). Its dimeric counterpart formed by subunits 3 and 4 also creates an identical interface to strengthen dimerization. Each subunit of BHMT donates its random loop region to intimately anchor one another at the center of the tetramer. The four subunits provide a relatively small interface area of 847 Å$^2$ for this oligomerization to occur (Fig 4B).

The interaction between the interface of subunits 1 and 2 is extensive. Specifically, Y38, H52, H252, E272, and the backbone oxygen of I357 of subunit 1 directly contact Q58, R65, R381, Y305, and N364 of subunit 2 to form hydrogen bonds (Fig 4C). In addition, E34, E51, D263, and R346 of subunit 1 interact with R278, K327, K340, and D371 of subunit 2 to form salt bridges to reinforce subunit–subunit interactions (Fig 4C). Residues H338 and W352 of subunit 1 specifically contact residues I262 and F269 of subunit 2, respectively, via hydrophobic interactions to further strengthen oligomerization (Fig 4C). The specific contacts between subunits 3 and 4 are identical to those of subunits 1 and 2, where hydrogen bonds and salt bridges are the predominant interactions.

At the center of the tetramer, each subunit of BHMT donates its random loop (residues 356–369) to anchor each other (Fig 4B). Particularly, each subunit of BHMT uses S366 and the backbone oxygen of P362 to make hydrogen bonds with its respective subunit (Fig 4D). Therefore, there are four identical pairs of hydrogen bonds formed among these four subunits. Residues that are involved in subunit–subunit interactions to strengthen the tetrameric oligomerization are shown in Fig 4D.

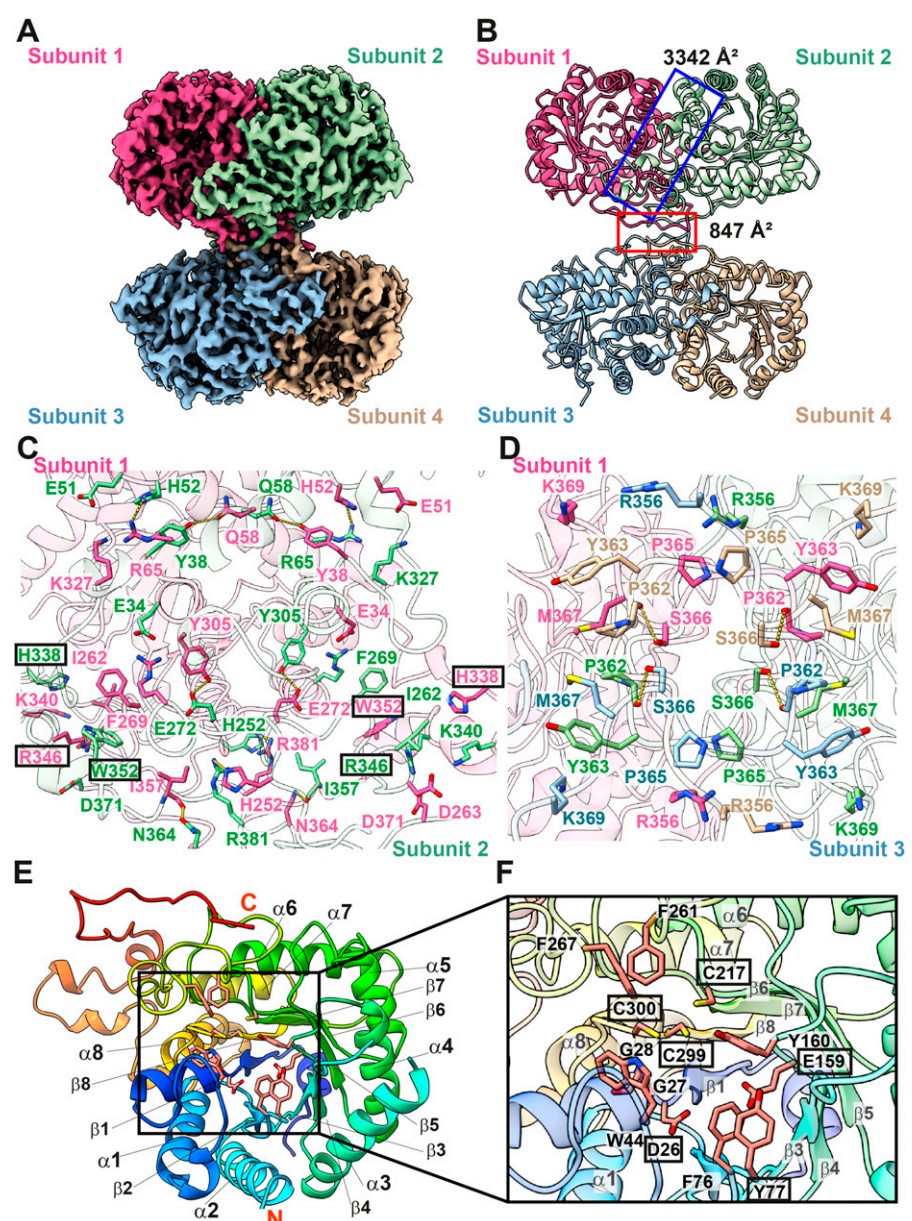

**Figure 4. Cryo-EM structure of human kidney BHMT.**
**(A)** Cryo-EM density map of tetrameric BHMT. The four protomers of BHMT are colored magenta, green, blue and wheat, respectively. **(B)** Ribbon diagram of the 2.62-Å resolution structure of tetrameric BHMT. The four protomers are colored magenta, green, blue, and wheat, respectively. The interface between subunits 1 and 2 covers an area of 3,342 Å$^2$, whereas the interface between subunits 1, 2, 3 and 4 possesses an area of 847 Å$^2$. **(C)** Composition of the interface between subunits 1 and 2. Residues involved in subunit–subunit interaction are in magenta (for subunit 1) and green (for subunit 2) sticks. Mutations on residues H338, R346, and W352 abolish the activity of the BHMT enzyme. These residues are highlighted with black rectangles. The secondary structural elements of subunit 1 and subunit 2 are colored pink and green. **(D)** Composition of the interface between subunits 1, 2, 3, and 4. Residues involved in subunit–subunit interaction are in magenta (for subunit 1), green (for subunit 2), blue (for subunit 3), and wheat (for subunit 4) sticks. The secondary structural elements of subunits 1, 2, 3, and 4 are colored pink, green, blue, and wheat, respectively. **(E)** A monomer of BHMT. The secondary structural elements are colored using a rainbow style (blue, N-terminus; red, C-terminus). Residues involved in forming the Zn$^{2+}$- and substrate-binding sites within the $\beta/\alpha$ barrel are in orange sticks. **(F)** The Zn$^{2+}$- and substrate-binding sites. Residues engaged in forming the Zn$^{2+}$- and substrate-binding sites are highlighted with orange sticks. Mutations on residues D26, E159, C217, C299, and C300 significantly diminish the activity of BHMT. These residues are highlighted with black rectangles. The secondary structural elements of BHMT are colored using a rainbow style (blue, N-terminus; red, C-terminus).

Alanine scanning mutagenesis suggests that H338, R346, and W352, located at the subunit–subunit interfaces of the human BHMT enzyme (Fig 4C), are critical residues (Szegedi & Garrow, 2004). Mutations of these individual residues abolished enzymatic activity, suggesting that subunit–subunit interactions may be a prerequisite for the function of this enzyme.

Each BHMT subunit establishes a $\beta/\alpha$ barrel consisting of eight $\beta$-strands and eight $\alpha$-helices. This $\beta/\alpha$ barrel houses a Zn-binding site and a substrate-binding site (Fig 4E). It has been reported that the three conserved cysteines, C217, C299, and C300 (Evans et al, 2002), and Y160 (González et al, 2004), are responsible for coordinating a Zn$^{2+}$ ion. In our cryo-EM structure of BHMT, these four residues are located near each other (Fig 4E and F). However, no Zn$^{2+}$ was found to bind within the vicinity, suggesting this conformation should represent the Zn$^{2+}$ free form. The crystal structure of S($\delta$-carboxybutyl)-L-homocysteine–bound BHMT indicates that the substrate was bound next to the Zn$^{2+}$-binding site, where D26, G27, G28, W44, F76, Y77, E159, Y160, F261, and F267 contribute to form this substrate-binding site (Evans et al, 2002). This structure also depicted that Zn$^{2+}$ was bound by C217, C299, and C300, but the sulfanyl atom of bound S($\delta$-carboxybutyl)-L-homocysteine participated in forming the fourth Zn$^{2+}$ coordination. In our cryo-EM structure, no extra density was observed within this substrate-binding site, indicating that there was no endogenous ligand present at this site (Fig 4F).

An experimental study using recombinant human liver BHMT suggested that the three cysteine residues, C217, C299, and C300 (Fig 4F), are critical for Zn$^{2+}$ binding. A mutation of any of these cysteines

results in complete loss of activity of this enzyme (Breksa & Garrow, 1999). Furthermore, using rat liver BHMT, it was found that mutations of residues corresponding to substrate-binding residues of the human enzyme, including D26, Y77, and E159 (Fig 4F), significantly depletes enzyme activity (González et al, 2003).

# Discussion

Cellular function is a complex event that involves a cluster of interactions among different proteins, enzymes, and even small molecules. Therefore, systems biology is a desirable approach to study living tissues and organs as it offers a comprehensive view of biological processes. We previously developed the BaR methodology for the purpose of elucidating high-resolution systems structural proteomics of raw biological samples. In the present work, we utilize BaR to study human kidney microsomes at near atomic resolutions. Through this approach, we simultaneously identified and solved high-resolution cryo-EM structures of four important kidney enzymes to resolutions between 2.62 and 2.88 Å. The presence of these enzymes in the kidney sample was also confirmed by proteomics using LC–MS/MS (Table S2).

It appears that these four kidney enzymes are critical components of several cellular processes, including protein biosynthesis and quality control, biosynthesis of retinoic acid, gluconeogenesis and glycolysis, and regulation and metabolism of amino acids. These enzymes are also directly associated with several disease states, notably cardiovascular disease and diabetes. Specifically, in type 2 diabetes, an increased concentration of homocysteine in the plasma has been identified as a significant risk factor. The BHMT enzyme plays a major role in the catabolism of homocysteine and is capable of lowering homocysteine levels (Wijekoon et al, 2007).

The positive effects of FPA have been well documented. Intriguingly, an administration of the recombinant enzyme FPA can reduce blood glucose levels, attenuate inflammation, and ameliorate type 1 diabetes in a murine model (Yan et al, 2020; Pirovich et al, 2021). It is possible that this approach may allow for the development of novel therapeutic strategies to combat this condition.

GANAB inhibitors are commonly used as hypoglycemic drugs. These inhibitors are recommended as potential first-line agents, particularly for newly diagnosed type 2 diabetic patients (Yang et al, 2014). These inhibitors have a significant effect on glycemic control and insulin levels (Van de Laar et al, 2005).

ALDH enzymes are rate-limiting enzymes that convert retinaldehyde to retinoic acid. These enzymes are directly related to the control of adipogenesis and energy homeostasis and are linked to several pathologies such as obesity, diabetes, and cardiovascular disease. Notably, ALDH1A1 knockdown can limit weight gain and improve glucose homeostasis in obese mice (Kiefer et al, 2012b). In addition, it has been observed that *ALDH1A1*-deficient mice displayed significantly decreased fasting glucose concentrations compared with WT controls (Kiefer et al, 2012a). These mice appear to be protected from diet-induced obesity and type 2 diabetes (Kiefer et al, 2012a). Therefore, ALDH1A1 is a potential drug target for the control of adipogenesis and energy homeostasis–related diseases.

The fact that all these enzymes are tightly connected to a single condition, diabetes, makes our BaR methodology exciting in that it enables us to study these proteins simultaneously in a single sample. As tissue and organ samples are inherently heterogenous and complex, our BaR approach can be used to overcome sample impurity and heterogeneity issues, enabling us to simultaneously solve structures of a variety of enzymes from a tissue sample at high resolutions. In this particular case, these four kidney enzymes can be further studied in the context of "healthy" versus "diabetic" samples, potentially uncovering differences that can be exploited for therapeutic benefit.

Although BaR allows us to handle raw sample and solve structures of different proteins from a single cryo-EM grid; this methodology has its limitations. For example, it is not easy to identify a particular protein if the population of the protein is <5% within the heterogeneous sample. In addition, it is difficult to unambiguously identify a protein when the cryo-EM map is lower than 3.5 Å in resolution. Further, preferential orientation of the protein images can be a problem, where BaR may not be able to overcome this specific issue. Despite these, it is expected that the BaR methodology can be a useful tool to help illuminate the details of biological pathways, networks, and even mechanisms of diseases at near-atomic resolution in the near future.

# Materials and Methods

### Human kidney microsome lysate

Human kidney microsomes were purchased from SEKISUI XenoTech. These microsomes were resuspended in buffer containing 20 mM Tris–HCl (pH 7.5), 100 mM NaCl, and 5 mM sodium cholate. Insoluble material from the microsome lysate was removed by ultracentrifugation at 20,000$g$ for 30 min. The extracted lysate was enriched using a Superdex 200 column (GE Healthcare) equilibrated with 20 mM Tris–HCl, pH 7.5, and 100 mM NaCl. Two major peaks located at 100–200 kD and 250–650 kD were isolated after the enrichment process. These two peaks were collected separately for single-particle cryo-EM imaging and mass spectrometry.

### Electron microscopy sample preparation and data collection

Human kidney microsome lysate samples were concentrated to 0.6 mg/ml, applied to glow-discharged holey carbon grids (Cu R1.2/1.3, 300 mesh; Quantifoil), blotted for 15 s, and plunge-frozen in liquid ethane using a Vitrobot (Thermo Fisher Scientific). The grids were then transferred into cartridges. A Titan Krios cryo-electron transmission microscope (Thermo Fisher Scientific) was used to collect cryo-EM images. The images were recorded at 1–2.5 μm defocus on a K3 direct electron detector (Gatan) using super-resolution and correlated-double sampling mode at nominal 81K magnification, corresponding to a sampling interval of 1.07 Å/pixel (super-resolution, 0.535 Å/pixel). Each micrograph was exposed for 3.5 s with 11.14 e-/A2/s dose rate (total specimen dose, 38.75 e-/A2). 52 frames were captured per specimen area using SerialEM (Mastronarde, 2005).

**Life Science Alliance**

### Data processing

Enrichment of the human microsome raw lysate was done using size exclusion chromatography with each peak processed separately. Super-resolution image stacks were aligned and binned by 2 using cryoSPARC (Punjani et al, 2017) to give a final pixel size of 1.07 Å/pixel. Contrast transfer functions were estimated using patch contrast transfer function in cryoSPARC (Punjani et al, 2017). A modified BaR protocol was used to determine the structures of multiple proteins for both cryo-EM experiments (Fig S1), closely after the initial BaR application that was used for cryo-EM data optimization in prokaryotic systems (Su et al, 2021). After manual inspection to discard poor images, all micrographs were picked using pre-trained models based on many publicly available datasets provided by Topaz (Bepler et al, 2019, 2020), giving rise to raw initial particle sets. These particles were classified with several rounds of 2D classification until no new classes were seen. Visually similar classes were then manually combined to determine initial low-resolution maps for individual target proteins. Iterative rounds of ab initio 3D reconstruction with 2D classes in cryoSPARC followed by homogeneous refinement were then performed to obtain low-resolution 3D models. Featureless particles, normally discarded, were used to build noisy ab initio 3D models. At this stage, many of the initial models suffered from orientation bias and low particle counts. To help improve overall model quality, both the low-resolution 3D models and the noisy ab initio 3D models were used as targets for three rounds of heterogeneous refinement of the raw initial particle sets, resulting in higher particle counts for each target protein. It was common for new 2D classes to appear after the initial round of BaR. These new, low-resolution classes were manually solved then used as additional inputs for heterogeneous refinement. This process was repeated until no new clear classes were uncovered from the remaining particles by either 2D or 3D classification. A final round of retrieval with all maps was used to separate particles for each dataset.

Individual particle subsets of each target protein were then re-extracted by Bin 1 and cleaned using multiple rounds of 2D and heterogeneous refinement with 3D ab initio models. C1 symmetry was applied to refine each structure with non-uniform refinement using cryoSPARC (Punjani et al, 2017). Symmetry was assayed in Chimera for cyclic, dihedral, tetrahedral, octahedral, and icosahedral symmetries in standard coordinate systems. If the correlation of the map with itself was more than 0.99, then the specific symmetry type was assigned. Non-uniform refinement was then performed with this symmetry parameter. For those maps that remained C1 symmetry, local refinement was followed without a distinct symmetry to obtain final maps.

The quality of the maps corresponding to each protein based on cryoSPARC (Punjani et al, 2017) was of high resolution, enabling us to identify the primary sequence of these proteins directly from the cryo-EM maps using DeepTracer (Pfab et al, 2021). The resolutions of these maps were also verified using RELION (Scheres, 2012). These protein sequences were then submitted to the NCBI protein blast (States & Gish, 1994) for protein identification, and their identities were confirmed by proteomics (Table S2).

### Model building and refinement

Model buildings of GANAB, ALDH1A1, FPA, and BHMT were based on the cryo-EM maps generated from the BaR methodology. The subsequent model rebuilding processes were performed using Coot (Emsley & Cowtan, 2004). Structure refinements were done using the phenix.real_space_refine program (Afonine et al, 2018) from the PHENIX suite (Adams et al, 2002). The final atomic models were evaluated using MolProbity (Chen et al, 2010). The statistics associated with data collection, 3D reconstruction, and model refinement are included in Table S1.

### LC–MS protein sample preparation and identification

Four µg of each peak of the collected kidney microsome lysate was separately denatured in buffer containing 50 mM $NH_4HCO_3$ and 8 M urea. 10 mM DTT (final concentration) was added to reduce the sample at 25°C for 30 min. The sample was then alkylated with 25 mM iodoacetamide at 25°C for 30 min. Each lysate sample was further diluted by four times using digestion buffer containing 100 mM $NH_4HCO_3$ and trypsin/Lys-C mix (1:20, enzyme:substrate). The samples were then digested overnight at 25°C. The digested peptides were desalted using a reverse-phase C18 Microspin column (Nest Group), washed twice with 150 µl aqueous solution containing 0.1% formic acid, and eluted with 150 µl aqueous solution containing 80% acetonitrile and 0.1% formic acid.

After digestion, LC–MS/MS was performed using the Thermo Scientific Fusion Lumos mass spectrometry system (Thermo Fisher Scientific). Tryptic peptides were loaded onto a Dionex 15 cm × 75 µm id Acclaim PepMap C18, 2 µm, 100 Å reversed-phase capillary chromatography column. The peptides were chromatographed with a linear gradient of acetonitrile from 2% to 35% in aqueous solution containing 0.1% formic acid for 90 min at a rate of 300 nl/min. The eluent was directly introduced into the mass spectrometer operated in data-dependent MS to MS/MS switching mode with the collision-induced dissociation mode. Full MS scanning was performed at 70,000 resolution between m/z = 350 and m/z = 1,500. Proteins were identified by comparing all of the experimental peptide MS/MS spectra with the UniProt human proteome database using a database search engine, MassMatrix (version 3.12). Carbamidomethylation of cysteine was set as a fixed modification, whereas variable modifications included oxidation of methionine to methionine sulfoxide and acetylation of N-terminal amino groups. For peptide/protein identification, a strict trypsin specificity was applied. The minimum peptide length was set to 6, the maximum missed cleavage was set to 2, and the cutoff false discovery rate was set to 0.025.

## Data Availability

Coordinates and EM maps for GANAB, ALDH1A1, FPA, and BHMT can be found at pdb accession numbers 8D43, 8D46, 8D44, and 8D45, and EMDB accession codes EMD-27173, EMD-27176, EMD-27174, and EMD-27175, respectively.

# Supplementary Information

# Acknowledgements

We thank Dr. Philip A Klenotic for proofreading the article. This work was supported by an NIH Grant R01AI145069 (EW Yu). The mass spectrometer used was purchased with an NIH Shared Instrument Grant (S10 RR031537). We thank Belinda Willard and Ling Li for the acquisition of mass spectrometry data. We are grateful to the Cryo-Electron Microscopy Core at the CWRU School of Medicine and Dr. Kunpeng Li for access to the sample preparation and Cryo-EM instrumentation.

## Author Contributions

M Lyu: conceptualization, data curation, formal analysis, validation, investigation, visualization, methodology, and writing—review and editing.
C-C Su: conceptualization, data curation, formal analysis, validation, and investigation.
M Miyagi: conceptualization, data curation, formal analysis, validation, and investigation.
EW Yu: conceptualization, data curation, formal analysis, supervision, funding acquisition, validation, investigation, visualization, methodology, project administration, and writing—original draft, review, and editing.

## Conflict of Interest Statement

The authors declare that they have no conflict of interest.

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
