## [Reviewer comments · Life Science Alliance]

Life Science Alliance

Simultaneous solving high-resolution structures of various enzymes from human kidney microsomes

Meinan Lyu, Chih-Chia Su, Masura Miyagi, and Edward Yu

DOI: <https://doi.org/10.26508/lsa.202201580>

Corresponding author(s): Edward Yu, Case Western Reserve University

Review Timeline:

Submission Date:	2022-06-27
Editorial Decision:	2022-08-08
Revision Received:	2022-10-11
Editorial Decision:	2022-10-26
Revision Received:	2022-11-07
Accepted:	2022-11-08

Transaction Report:

August 8, 2022

Re: Life Science Alliance manuscript #LSA-2022-01580-T

Prof. Edward W. Yu
Case Western Reserve University
Department of Pharmacology
Ohio 44106

Dear Dr. Yu,

Thank you for submitting your manuscript entitled "Simultaneous determination of high-resolution structures of a variety of enzymes from human kidney microsomes" to Life Science Alliance. The manuscript was assessed by expert reviewers, whose comments are appended to this letter. We invite you to submit a revised manuscript addressing the Reviewer comments.

Thank you for this interesting contribution to Life Science Alliance. We are looking forward to receiving your revised manuscript.

Sincerely,

B. MANUSCRIPT ORGANIZATION AND FORMATTING:

Reviewer #1 (Comments to the Authors (Required)):

This manuscript by Lyu et al. describes the implementation of the Build and Retrieve (BaR) methodology to study proteomics of human kidney microsomes. The authors have identified and determined the structure of four different enzymes - the heterodimeric complex glucosidase II (GANAB), retinaldehyde dehydrogenase (ALDH1A1), fructose-bisphosphate adolase (FPA) and betaine-homocysteine methyltransferase (BHMT) - using cryogenic electron microscopy combined with mass spectrometry. These enzymes are directly associated with chronic diseases including diabetes and cardiovascular diseases. Overall, the authors provided comprehensive structural analyses of each protein. Also, this work introduces an integrated approach between proteomics and structural biology which can be useful for studying other endogenous biological samples. I believe that this work is comprehensive and could be a good fit for Life Science Alliance. I have a couple of suggestions/questions for the authors to address:

- The two major peaks on the size exclusion chromatography should be shown in the supplementary figure.
- Each structure reveals some key residues that are involved either in stabilizing the interaction between subunits in the oligomerization, or in substrate binding. However, I would like to see more discussion on what these structures tell us in the context of disease-causing mutations, and how structural information translates into understanding the mechanism of human diseases. Mutations in these particular proteins associating with critical diseases in human health should also be mapped on to the structures.
- It would be great if the authors could share biochemical evidence or references to validate the interaction between the subunits and the substrate/ion binding sites in these enzymes.
- In the structure of GANAB, N97 is likely to be glycosylated. Has any biochemical or functional assay been performed on this residue previously?
- The authors state that the cryo-EM structure is similar to the crystal structure of the mouse homologous enzyme alpha-glucosidase II. The authors should include RMSD, superimposition and sequence similarity between the human and the mouse structures.
- In the GANAB structure, two hexacoordinated Ca²⁺ ions are found within the beta-subunit. What is the relevance of Ca²⁺ ions in their enzymatic activity? Has there been any mutagenesis work on the residues that coordinate Ca²⁺ ions?
- In the processing of the ALDH1A1 dataset, have the authors tried 3D classification or focus refinement in the cofactor-binding site region to attempt to locate an extra density for the NAD⁺ cofactor?
- In the processing of FPA, have the authors observed any complexes with this enzyme?
- The overall quality of each map is good. However, there seems to be some stretching of the density in the BHMT structure. I recommend including a Euler angle distribution of all particles used in the final map reconstruction in one of the supplementary figures.

Reviewer #2 (Comments to the Authors (Required)):

1. Lyu et al present a concise and beautiful cryo-EM study identifying and visualizing 4 proteins from raw kidney microsomal lysate, GANAB, ALDH1A1, FPA, and BHSMT, using the previously-described cryo-EM workflow of Build-and-Retrieve, which includes preparation of cell lysate, protein isolation by SEC, cryo-EM analysis and model building. This work is a testament to the ever-more challenging nature of targets of cryo-EM structural characterization and represents the trajectory of not just visualizing, but also identifying critical cellular complexes and elucidating their functions by cryo-EM.
2. I have no demands or suggestions for additional experiments, merely comments that I believe, if addressed, would strengthen

the manuscript.

Major comments

- This brief and descriptive study relies heavily on work presented by Su et al in Nature Methods. As a result, the manuscript is highly derivative in its nature. Due to the technical aspects of this study, more in-depth methodological details should be included (rather than referenced to in the Su et al).
- It is unclear to me how the SEC run could result in only two peaks that spanned 100-200, and 250-650kDa, respectively; that's the entire separation span of the column. Do they have two macromolecular complexes or is this due to lack of further separation?
- It's unclear to me how particle picking was performed; where all particles in each sample (containing particles ranging in size) picked on were parameters used to only pick certain particles?
- Initial versus final 2D should be shown to illustrate particle heterogeneity and provide further insight into the BnR process.
- Is it unclear exactly how the symmetry was determined in the different maps.

Minor

- EM density maps are referred to as "high-quality", a more adequate description is "high-resolution".
- Were the maps validated using other programs, e.g. Relion?

3. I would like to see data from the sample preparation presented.

Reviewer #3 (Comments to the Authors (Required)):

The present manuscript follows a publication by the same group last year (Su et al, 2021). In that publication, the authors developed the "Build and Retrieve (BaR)" method, which enables the determination of cryo-EM structures of multiple proteins simultaneously from a heterogeneous protein sample. In this manuscript, utilizing BaR method, the authors solved high resolution cryo-EM structures of four kidney enzymes at one time from raw human kidney microsomal lysate. BaR method is indeed a useful tool to help researchers overcome homogeneity and purity problems of sample preparation. My major concern is the "efficiency" of this method since we do not know what protein structure will be solved until the last moment. For example, among the four structures solved by authors, the structures of human FPA and BHMT have been determined by crystallization previously. It is worthwhile to discuss the possible limitations of the BaR method and potential solution, if there is, in the discussion section. In sum, this work proved the feasibility of solving the atomic resolution structures of proteins from raw samples, I recommend publication after the below points are addressed.

1. Show gel filtration chromatography and SDS-PAGE of the purified samples in supplementary figure, label the fractions which applied to cryo-EM. The authors should highlight the bands or areas of the four proteins on SDS-PAGE gel to help readers visualize the proportion of target proteins in the mixture.

2. I would appreciate it if the box in the figure which used to show a close-up view can be considerably obvious.

Reviewer #1 (Comments to the Authors (Required)):

This manuscript by Lyu et al. describes the implementation of the Build and Retrieve (BaR) methodology to study proteomics of human kidney microsomes. The authors have identified and determined the structure of four different enzymes - the heterodimeric complex glucosidase II (GANAB), retinaldehyde dehydrogenase (ALDH1A1), fructose-bisphosphate adolase (FPA) and betaine-homocysteine methyltransferase (BHMT) - using cryogenic electron microscopy combined with mass spectrometry. These enzymes are directly associated with chronic diseases including diabetes and cardiovascular diseases. Overall, the authors provided comprehensive structural analyses of each protein. Also, this work introduces an integrated approach between proteomics and structural biology which can be useful for studying other endogenous biological samples. I believe that this work is comprehensive and could be a good fit for Life Science Alliance. I have a couple of suggestions/questions for the authors to address:

We really appreciate this reviewer's recommendation, indicating that "this work is comprehensive and could be a good fit for Life Science Alliance".

- The two major peaks on the size exclusion chromatography should be shown in the supplementary figure.

A new supplementary figure (Fig. S1) showing the two peaks from size exclusion chromatography has been included in this revised manuscript.

- Each structure reveals some key residues that are involved either in stabilizing the interaction between subunits in the oligomerization, or in substrate binding. However, I would like to see more discussion on what these structures tell us in the context of disease-causing mutations, and how structural information translates into understanding the mechanism of human diseases. Mutations in these particular proteins associating with critical diseases in human health should also be mapped on to the structures.

We have included several paragraphs, such as "These residues at the subunit-subunit interface could be very important for the function of GANAB, where a mutation of one or more of these residues could lead to devastating illnesses. For example, a missense mutation R817W has been identified in three patients with autosomal-dominant polycystic liver disease (Porath *et al*, 2016). Based on our cryo-EM structure, R817 is critical for stabilizing the interaction between the α and β subunits, as this α -subunit residue directly contacts N93 of the β -subunit to form a hydrogen bond (Figure 1D)." (p. 7), "These subunit-subunit interface residues are likely important for the function this enzyme. Interestingly, two missense mutations, A151S and I157T, have been identified for the human ALDH1A2 enzyme (Christy & Doss, 2015). These two mutations are strongly associated with congenital heart disease. The corresponding two amino acids in ALDH1A1 are the interface residues A134 and I140 (Figure 2C), where they may be critical for the tetrameric oligomerization of this enzyme." (p. 9), "Hereditary fructose intolerance is an autosomal recessive disease caused by the catalytic deficiency of FPA. Several missense mutations have been identified that are directly linked to this disease (Tolan, 1995). One of these missense mutants is the conversion of L257 to a proline (Tolan, 1995). This residue is located at the subunit-subunit interface to strengthen oligomerization (Figure 3C)." (p. 12), and "Alanine

scanning mutagenesis suggests that H338, R346 and W352, located at the subunit-subunit interfaces of the human BHMT enzyme (Figure 4C), are critical residues (Szegeedi & Garrow, 2004) Mutations of these individual residues abolished enzymatic activity, suggesting that subunit-subunit interactions may be a prerequisite for the function of this enzyme.” (p. 14) in this revised manuscript to highlight the importance of these residues at subunit-subunit interfaces of these enzymes. Mutations of these interface residues could cause devastating diseases.

- It would be great if the authors could share biochemical evidence or references to validate the interaction between the subunits and the substrate/ion binding sites in these enzymes.

In this revised version of the manuscript, we have included several paragraphs, including “Interestingly, a study of the homologous α -glucosidase II enzyme from *Schizosaccharomyces pombe* using alanine scanning mutagenesis indicated that mutations of residues E73 and E114 of the β -subunit inactivated its function. This work underscored the importance of these glutamates to coordinate with Ca^{2+} ions. The two corresponding residues in the human kidney GANAB enzyme are E64 and E105 which are also responsible for anchoring Ca^{2+} ions (Figure 1F).” (p. 7), “ALDH1A1 has been found to effectively increase NADH levels and promote tumor growth. Residues involved in creating the NAD^+ -binding site are presumed to be critical for the function of this enzyme. Indeed, it has been observed that cells harboring a mutation of the conserved interface residue K193 (K193Q or K193R) of ALDH1A1 are much less tumorigenic when compared with cells carrying the wild-type ALDH1A1 enzyme (Liu *et al*, 2021). Further, another mutagenesis study indicated that the activity of ALDH1A1 was significantly reduced when two glycine residues located at the NAD^+ -binding site, G246 and G251, were replaced by alanines (Wang *et al*, 2017), probably due to the effect of steric hindrance.” (p. 10), “Several studies have shown that the glutamate and lysine residues within the substrate-binding site are critical for the function of FPA enzymes (Gupta *et al*, 1993; Hartman & Brown, 1976; Lai *et al*, 1974; Lobb *et al*, 1975; Morris & Tolan, 1993, 1994). Interestingly, a patient with hereditary fructose intolerance was found to have a six-nucleotide deletion in exon 6. This deletion perturbs the position and orientation of the corresponding K147 and E188 residues of human FPA at the active site (Figure 3F) (Santamaria *et al*, 1999). In addition, a separate study depicted that R304 (Figure 3F) changed to a tryptophan at this catalytic site gives rise to hereditary fructose intolerance (Tolan, 1995). This missense mutation probably diminishes the binding of substrates and leads to this disease.” (p. 12), and “Interestingly, an experimental study using recombinant human liver BHMT suggested that the three cysteine residues, C217, C299 and C300 (Figure 4F), are critical for Zn^{2+} binding. A mutation of any of these cysteines results in complete loss of activity of this enzyme (Brekša & Garrow, 1999). Further, using rat liver BHMT, it was found that mutations of residues corresponding to substrate-binding residues of the human enzyme, including D26, Y77 and E159 (Figure 4F), significantly depletes enzyme activity (González *et al*, 2003).” (p. 15) to highlight the importance of residues located at the binding sites of these enzymes.

- In the structure of GANAB, N97 is likely to be glycosylated. Has any biochemical or functional assay been performed on this residue previously?

Yes, it has been found that GANAB is glycosylated. A few statements “This observation is indeed in good agreement with results from a biochemical study that determined that human

GANAB is glycosylated (Martiniuk *et al*, 1985). Interestingly, the x-ray structure of mouse α -glucosidase II also depicts that this enzyme is glycosylated at residue N97 with the same elongated glycan chain (Caputo *et al*, 2016).” have been included in this revised version of the manuscript (p. 6) to address this.

- The authors state that the cryo-EM structure is similar to the crystal structure of the mouse homologous enzyme alpha-glucosidase II. The authors should include RMSD, superimposition and sequence similarity between the human and the mouse structures.

We thank this reviewer for this constructive comment. A new paragraph “ Our cryo-EM structure indicates that human kidney GANAB is composed of one α -subunit and one β -subunit (Figure 1A), which is in good agreement with the crystal structure of the homologous mouse enzyme α -glucosidase II (Caputo *et al*, 2016). Protein sequence alignment shows that these two enzymes share 96% protein sequence similarity. Superimposition of the cryo-EM structure of the human enzyme to the x-ray structure of the mouse enzyme (PDB ID: 5F0E) (Caputo *et al*, 2016) gives rise to a root-mean-square-deviation (r.m.s.d.) of 0.52 Å (for 850 C α atoms).” has been added on p. 5 of this revised manuscript to address this comment.

- In the GANAB structure, two hexacoordinated Ca²⁺ ions are found within the beta-subunit. What is the relevance of Ca²⁺ ions in their enzymatic activity? Has there been any mutagenesis work on the residues that coordinate Ca²⁺ ions?

Yes, a mutagenesis study on a homologous enzyme from *Schizosaccharomyces pombe* has been done, indicating that the glutamate residues responsible for anchoring bound Ca²⁺ ions are important for the function. A statement “Interestingly, a study of the homologous α -glucosidase II enzyme from *Schizosaccharomyces pombe* using alanine scanning mutagenesis indicated that mutations of residues E73 and E114 of the β -subunit inactivated its function. This work underscored the importance of these glutamates to coordinate with Ca²⁺ ions. The two corresponding residues in the human kidney GANAB enzyme are E64 and E105 which are also responsible for anchoring Ca²⁺ ions (Figure 1F).” has been added on p.7 of the revised manuscript to address this.

- In the processing of the ALDH1A1 dataset, have the authors tried 3D classification or focus refinement in the cofactor-binding site region to attempt to locate an extra density for the NAD⁺ cofactor?

Yes, we tried both the 3D classification and focused refinement at the cofactor-binding site region. We did not observe any extra densities corresponding to the bound NAD⁺ cofactor.

- In the processing of FPA, have the authors observed any complexes with this enzyme?

We also tried both the 3D classification and focused refinement at the substrate-binding site. Again, we did not observe any extra densities corresponding to bound substrate.

- The overall quality of each map is good. However, there seems to be some stretching of the density in the BHMT structure. I recommend including a Euler angle distribution of all particles

used in the final map reconstruction in one of the supplementary figures.

We have included Euler angle distributions for all structures in the supplementary figures (Figs. S3-S6)

Reviewer #2 (Comments to the Authors (Required)):

1. Lyu et al present a concise and beautiful cryo-EM study identifying and visualizing 4 proteins from raw kidney microsomal lysate, GANAB, ALDH1A1, FPA, and BHSMT, using the previously-described cryo-EM workflow of Build-and-Retrieve, which includes preparation of cell lysate, protein isolation by SEC, cryo-EM analysis and model building. This work is a testament to the ever-more challenging nature of targets of cryo-EM structural characterization and represents the trajectory of not just visualizing, but also identifying critical cellular complexes and elucidating their functions by cryo-EM.

We thank this reviewer's glowing comment, indicating that this is "a concise and beautiful cryo-EM study" and "a testament to the ever-more challenging nature of targets of cryo-EM structural characterization and represents the trajectory of not just visualizing, but also identifying critical cellular complexes and elucidating their functions by cryo-EM".

2. I have no demands or suggestions for additional experiments, merely comments that I believe, if addressed, would strengthen the manuscript.

Major comments

-This brief and descriptive study relies heavily on work presented by Su et al in Nature Methods. As a result, the manuscript is highly derivative in its nature. Due to the technical aspects of this study, more in-depth methodological details should be included (rather than referenced to in the Su et al).

The Build-and-Retrieve procedures, starting from initial 2D classification to final map construction, have been rewritten in the "Method" section of this revised manuscript to describe the details of the protocol, following the reviewer's comment.

-It is unclear to me how the SEC run could result in only two peaks that spanned 100-200, and 250-650kDa, respectively; that's the entire separation span of the column. Do they have two macromolecular complexes or is this due to lack of further separation?

Our goal for using SEC is to enrich protein particles, allowing us to obtain more particle counts. We only focus on particles with size ≥ 100 kDa, which would make it easier for high-resolution cryo-EM structural determination. During the enrichment process, we were also able to separate the large peak corresponding to aggregation from protein particles. For size ≥ 100 kDa, we only observed two peaks, 100-200 kDa and 250-650 kDa, from our sample (see Fig. S1).

-It's unclear to me how particle picking was performed; where all particles in each sample

(containing particles ranging in size) picked on were parameters used to only pick certain particles?

The initial particle stack was picked by using Topaz (Bepler *et al*, 2019, 2020) with the default ResNet16 (64 units) pretrained model (which has been trained on a large corpus of datasets including a wide variety of particles).

-Initial versus final 2D should be shown to illustrate particle heterogeneity and provide further insight into the BnR process.

The initial 2D images have been included in Fig. S2 (Build and Retrieve workflow) of the supplementary materials of this revised manuscript. The selected particles of GANAB, ALDH1A1, FPA, and BHMT are highlighted with magenta hexagons, yellow squares, pink circles, and green stars, respectively, in this supplementary figure. The final 2D images of these enzymes are included in Figs. S3B, S4B, S5B and S6B.

-Is it unclear exactly how the symmetry was determined in the different maps.

We first applied C1 symmetry to each protein. We then determined the symmetry using Chimera. The criterion is that the correlation of the maps before and after a particular symmetry transformation has to be ≥ 0.99 to ensure that the symmetry is a correct.

Minor

-EM density maps are referred to as "high-quality", a more adequate description is "high-resolution".

The use of "high-quality" to describe cryo-EM maps has been switched to "high-resolution", following this reviewer's suggestion.

-Were the maps validated using other programs, e.g. Relion?

Yes, the resolutions of the maps were validated using RELION. For example, the resolutions of the maps corresponding to GANAB, ALDH1A1, FPA and BHMT were reported 2.88 Å, 2.84 Å, 2.80 Å and 2.62 Å, respectively, in cryoSPARC. These resolutions became 2.89 Å, 2.86 Å, 2.84 Å and 2.65 Å, respectively, from RELION. A statement "The resolutions of these maps were also verified using RELION (Scheres, 2012)" has been added in the method section of this revised manuscript.

3. I would like to see data from the sample preparation presented.

In this revised manuscript, we have included a new supplementary figure (Fig. S1), showing the SEC peaks for sample enrichment and SDS-PAGE of these two SEC peaks.

Reviewer #3 (Comments to the Authors (Required)):

The present manuscript follows a publication by the same group last year (Su et al, 2021). In that publication, the authors developed the "Build and Retrieve (BaR)" method, which enables the determination of cryo-EM structures of multiple proteins simultaneously from a heterogeneous protein sample. In this manuscript, utilizing BaR method, the authors solved high resolution cryo-EM structures of four kidney enzymes at one time from raw human kidney microsomal lysate. BaR method is indeed a useful tool to help researchers overcome homogeneity and purity problems of sample preparation. My major concern is the "efficiency" of this method since we do not know what protein structure will be solved until the last moment. For example, among the four structures solved by authors, the structures of human FPA and BHMT have been determined by crystallization previously. It is worthwhile to discuss the possible limitations of the BaR method and potential solution, if there is, in the discussion section. In sum, this work proved the feasibility of solving the atomic resolution structures of proteins from raw samples, I recommend publication after the below points are addressed.

We very much appreciate this reviewer's recommendation for the possibility of publishing our manuscript in Life Science Alliance. A discussion regarding possible limitations of the BaR methodology has been included in the last paragraph of the "Discussion" section of this revised version of the manuscript.

1. Show gel filtration chromatography and SDS-PAGE of the purified samples in supplementary figure, label the fractions which applied to cryo-EM. The authors should highlight the bands or areas of the four proteins on SDS-PAGE gel to help readers visualize the proportion of target proteins in the mixture.

A new supplementary figure (Fig. S1), showing the two SEC peaks and their corresponding SDS-PAGE, has been added in this revised version of the manuscript, following this reviewer's suggestion.

2. I would appreciate it if the box in the figure which used to show a close-up view can be considerably obvious.

We thank for this reviewer's suggestion. The boxes in the figures have been remade, making them easier to see.

October 26, 2022

RE: Life Science Alliance Manuscript #LSA-2022-01580-TR

Prof. Edward W. Yu
Case Western Reserve University
Department of Pharmacology
Cleveland, Ohio 44106

Dear Dr. Yu,

Thank you for submitting your revised manuscript entitled "Simultaneous solving high-resolution structures of various enzymes from human kidney microsomes". We would be happy to publish your paper in Life Science Alliance pending final revisions necessary to meet our formatting guidelines.

- please address Reviewer 1's final comments
- please upload your supplementary figure files as single files and add your supplementary figure files to the main manuscript text
- please add the Twitter handle of your host institute/organization as well as your own or/and one of the authors in our system
- please add the author contributions and a conflict of interest statement to the main manuscript text
- please use the [10 author names, et al.] format in your references (i.e. limit the author names to the first 10)
- please upload your table files as separate editable excel or doc files or make sure that they're included in the doc file of your manuscript
- the pdb accession numbers and EMBD accession codes should now be made publicly accessible

A. FINAL FILES:

B. MANUSCRIPT ORGANIZATION AND FORMATTING:

Sincerely,

Reviewer #1 (Comments to the Authors (Required)):

The revised manuscript by Lyu et al., has been improved by addressing the major concerns raised in previous review of the original manuscript. Also, the figures are improved and easy to follow. The authors have provided comprehensive cryo-EM structural analyses of four important enzymes isolated from human kidney microsomes in context of human diseases. The Build and Retrieve approach has proven to be an effective tool for studying challenging targets from raw biological samples with a significant potential to advance the field of structural biology. I remain supportive for the work to be published in Life Science Alliance after the minor issues to be addressed.

In page 16, the paragraph contains a repetition 'of protein biosynthesis and quality control...'.

The word 'Interestingly' appears too frequently, and can be omitted or substituted with other words.

-please add a conflict of interest statement to the main manuscript text

The conflict of interest statement has been included in this revised manuscript.

-the pdb accession numbers and EMDB accession codes should now be made publicly accessible

We have requested to release all PDBs and EMDBs and made them available to the public.

November 8, 2022

RE: Life Science Alliance Manuscript #LSA-2022-01580-TRR

Prof. Edward W. Yu
Case Western Reserve University
Department of Pharmacology
Cleveland, Ohio 44106

Dear Dr. Yu,

Thank you for submitting your Research Article entitled "Simultaneous solving high-resolution structures of various enzymes from human kidney microsomes". It is a pleasure to let you know that your manuscript is now accepted for publication in Life Science Alliance. Congratulations on this interesting work.

DISTRIBUTION OF MATERIALS:

Again, congratulations on a very nice paper. I hope you found the review process to be constructive and are pleased with how the manuscript was handled editorially. We look forward to future exciting submissions from your lab.

Sincerely,
